# Social and Behavioral Predictors of Adolescents’ Positive Attitude towards Life and Self

**DOI:** 10.3390/ijerph16224404

**Published:** 2019-11-11

**Authors:** Marta Malinowska-Cieślik, Joanna Mazur, Hanna Nałęcz, Agnieszka Małkowska-Szkutnik

**Affiliations:** 1Department of Environmental Health, Faculty of Health Sciences, Jagiellonian University, Medical College, 20 Grzegorzecka Str., 31-531 Krakow, Poland; 2Collegium Medicum, University of Zielona Gora, 26 Zyty Str., 65-046 Zielona Gora, Poland; 3Department of Child and Adolescent Health, Institute of Mother and Child, 17A Kasprzaka Str., 01-211 Warsaw, Poland; 4Faculty of Education, University of Warsaw, 16/20 Mokotowska Str., 00-561 Warsaw, Poland

**Keywords:** adolescents, positive attitude, health behaviors, social relations, neighborhood, school performance

## Abstract

Positive attitude is an important cognitive component of optimism. Although optimism has been widely studied in adolescents’ health, there is limited knowledge about social and behavioral determinants of their positive attitude. The aim of this study was to identify the main predictors of a positive attitude towards life and self in adolescence. Data were collected in 2010 from 2562 Polish adolescents, aged 15 to 17 years old, within the Health Behavior in School-Aged Children survey. A positive attitude was measured using the 4-item Positive Attitude Scale (PAS). Univariate analysis of variance was conducted and then hierarchical linear regression models adjusted for gender, age and family affluence were estimated. The mean PAS score was 13.25 (SD = 3.74), on the scale ranged 0–20. Eight out of 18 variables were included in the final model, which explained 25.1% of PAS variability. Communication in the family and with peers, as well as neighborhood social capital showed the strongest impact on positive attitude in adolescents. Physical activity, eating breakfast and school performance were also found to be important predictors. The results of the study highlight the need to include the development of interpersonal competences, promoting physical activity and supporting school performance, in adolescents’ mental health promotion programs, particularly in girls.

## 1. Introduction

Adolescence is an exceptionally plastic period of life, in which personality traits and interaction with family, peers and the community are considered as significant aspects of development. Studies of adolescent health and well-being pay high attention to positive development and the enhancement of individual traits and home, school and community environments [1,2]. These studies analyzed personal assets such as resilience, self-esteem, self-efficacy, sense of coherence, and internal locus of control as influencing mental health and subjective well-being in adolescents. Adolescents’ psychological well-being is defined in relation to youth positive development and includes self-acceptance, positive intrapersonal relationships, autonomy and life satisfaction [3]. In regard to adolescents’ mental health, the longitudinal study of Chen and Harris [4] showed that positive family relationships are associated with better mental health and reduced depressive symptoms among females and males from early adolescence to midlife. Studies among Spanish young adolescents found a relationship between family socioeconomic status and physical and social well-being [5]. The World Health Organization (WHO) defines mental health as “a state of well-being in which every individual realizes his or her own potential, can cope with the normal stresses of life, can work productively and fruitfully, and is able to make a contribution to her or his community.” [6]. Therefore, monitoring and surveillance of the population’s mental health should include positive indicators of mental health. In many countries, national mental health promotion strategies consider such positive measures. For example, Canada’s strategy for mental health contains determinants organized in four domains at the individual, family and interpersonal, community, and societal levels [7]. The individual-level includes personality traits such as self-esteem, sense of mastery, sense of coherence, optimism and emotional intelligence. A positive attitude towards oneself and life is related to optimism, which in turn is associated with a sense of self-confidence, better coping with stress and stronger resilience [8,9]. Optimism protects and enhances mental health and health-related lifestyle changes [10,11]. This study investigates associations and the influence of social relations and health behaviors with adolescents’ positive attitude, which is related to optimism. Optimism is defined as a global expectation that more good, desirable things, rather than bad, undesirable things will happen in the future. Optimism has implications for physical and psychological well-being and a number of positive health-related outcomes [12]. The Life Orientation Test (LOT and LOT-Revised), a tool to assess dispositional optimism, developed by Scheier and Carver [13,14], has been used in many studies on health impacts. Many studies, including meta-analytic reviews, showed that optimism is associated with taking proactive steps to protect one’s health and has an impact on physical and psychological well-being, and is a significant predictor of positive physical health outcomes in adults [15,16,17]. In adolescents, optimism is also the stronger predictor of their mental health in comparison with health behaviors [18]. Other studies of youth health show that positive attributes and optimism are positively correlated with physical health and reduce metabolic and cardiometabolic risk in young people [19,20]. In regard to social support, dispositional optimism mediates adolescents’ perceived support from parents, peers, and teachers to psychological well-being [21]. In Japanese college students, optimism was positively correlated with perceived social support [22]. Oberle et al. [23] found that adolescents’ optimism predicted peers’ acceptance, in a gender-specific manner. Girls’ acceptance of peers was significantly predicted by higher levels of optimism.

In regard to school performance, the literature review by Suldo et al. [24] showed a bidirectional relationship between mental health and academic functioning and suggested that changes in one domain can predict changes in the other. Boman and Yates [25] showed that adolescents’ optimism was significantly related to their classroom involvement and to students’ self-reported adjustment and attitudes towards their school. Moor et al. [26] showed that the relationship between educational inequalities and life satisfaction is mediated by health-related behaviors among both genders, but to a greater extent in girls.

Regarding the above-mentioned studies describing the relationship between social relations, health behaviors, school performance, and adolescents’ optimism, it seems to be important to incorporate an analysis of social and behavioral determinants of a positive attitude, which contribute to the development of dispositional optimism. We assume that communication in the family, relations with peers, neighborhood social capital, health-enhancing, and health risk behaviors, academic achievements and school-related stress influence adolescents’ positive attitude. However, we should consider the reverse associations, which were proven in the above-mentioned studies [10,15,16,22,23,25].

We sought answers to the following questions:Are there significant relationships between health behaviors with adolescents’ positive attitudes towards themselves and their life?Do these associations persist, considering the influence of social factors and school performance?Which of the health-related behaviors impact adolescents’ positive attitude?Are positive associations with selected factors gender specific?

## 2. Materials and Methods

### 2.1. Study Design and Participants

The presented study is based on data from the Health Behavior in School-Aged Children survey, the World Health Organization Regional Office for Europe collaborative cross-national survey (HBSC) conducted in 2010 in Poland. The sample selection and organization of the school-based survey was done according to the international research protocol [27]. In Poland, the HBSC national survey cluster sampling of school classes were carried out. The primary sampling unit was schools, and a single class was chosen to be included in the sample. All pupils within the selected classes were included in the sample [28]. The approval of the study was granted by the Bioethical Committee operating at the Institute of Mother and Child, Warsaw, Poland, decision no: 16/2009 dated 16 October 2009. Standardized information with agreements to participate in the study was provided to parents and students. Anonymous, standard questionnaires were administered to 15- and 17-year-old pupils in a nationally representative sample of adolescents. Data consisted of 1551 responses from 15-year-old students (mean age 15.73; SD = 0.30) and 1411 from 17-year-old students (mean age 17.74; SD = 0.31). The response rate was 89.3% [28]. Residents of large cities, small towns, and villages constituted respectively 34.6%, 26.5%, and 38.9% of all respondents. According to family affluence, 24.9%, 43.6%, and 31.5% of respondents belonged to relatively poor, average, and rather well-off families.

The presented study is based on data from the Health Behavior in School-Aged Children survey, the World Health Organization Regional Office for Europe collaborative cross-national survey (HBSC) conducted in 2010 in Poland. The sample selection and organization of the school-based survey was done according to the international research protocol [27]. In Poland, the HBSC national survey cluster sampling of school classes were carried out. The primary sampling unit was schools, and a single class was chosen to be included in the sample. All pupils within the selected classes were included in the sample [28]. The approval of the study was granted by the Bioethical Committee operating at the Institute of Mother and Child, Warsaw, Poland, decision no: 16/2009 dated 16 October 2009. Standardized information with agreements to participate in the study was provided to parents and students. Anonymous, standard questionnaires were administered to 15- and 17-year-old pupils in a nationally representative sample of adolescents. Data consisted of 1551 responses from 15-year-old students (mean age 15.73; SD = 0.30) and 1411 from 17-year-old students (mean age 17.74; SD = 0.31). The response rate was 89.3% [28]. Residents of large cities, small towns, and villages constituted respectively 34.6%, 26.5%, and 38.9% of all respondents. According to family affluence, 24.9%, 43.6%, and 31.5% of respondents belonged to relatively poor, average, and rather well-off families.

### 2.2. Instruments

The Positive Attitude Scale (PAS), developed by Mazur et al. [29] on the basis of review and pilot testing of four instruments measuring resilience, was the main outcome measure. It allows a single factor to be identified, which can be interpreted as a positive attitude towards oneself and life. Similar items could be found in the Wagnild and Young Resilience Scale [30]. Polish adaption of the full 14-item Wagnild and Young Scale has been recently developed and published by Surzykiewicz et al. [31].

The following positively worded items are included in the PAS:-I am proud of my achievements.-I always find a reason to be happy.-My belief in myself gets me through difficult times in my life.-My life has meaning.

The respondents were asked to assess whether the given statement describes them accurately or not, using six categories of responses, without a neutral category, from 0—it describes me very inaccurately—to 5—it describes me very well. Based on these four questions, a crude (0–20) and a standardized z-score index were developed (mean = 0; SD = 1). The first one was used in descriptive analyses, while the second one in the regression model. Higher scores are interpreted as a positive result. Missing values constituted 1.4% and were replaced by the conditional mean estimated for individuals who responded identically to the remaining PAS questions. In the study sample, the positive psychometric properties of the PAS were confirmed. The reliability of the PAS, evaluated using Cronbach’s alpha, was 0.815.

A total of 17 potential positive attitude predictors, grouped into categories, were analyzed: sociodemographic variables, health-related behaviors, school performance, and social relations. The characteristics of the sample in terms of these variables is shown in Table 1 and Table 2.

Family affluence was measured using the Family Affluence Scale (FAS), which included four questions about the number of computers and cars in the family, the student’s own room, and holiday trips with the family outside the place of residence in the preceding year. The FAS is treated as a quasi-continuous scale (0–7 points); it is a standard tool in the HBSC survey, used in analyzing social inequalities in health [32].

It was assumed that the student’s functioning at school was related to the performance of developmental tasks, and it is generally a source of stress. A single question on school achievements was included (0—below average—to 3 – very good) along with the scale of perceived school-related stress. The students used a five-point scale to identify how much they agree with four statements, e.g., I find studying difficult (0—strongly disagree—to 4—strongly agree). The reliability of that scale was good in the examined sample (Cronbach’s alpha = 0.742).

Social relations were examined based on two communication scales relating to family and peers and the assessment of the social capital of the neighborhood. The scale of clarity of family communication derives from a larger research tool: Family Dynamics Measure II (FDM II) by Barnhill [33]. The responses (indicating the extent to which respondents agree) were encoded to 0‒5 points respectively, and a higher score indicated a positive state. Cronbach’s alpha for this scale was 0.874. The peer communication scale was taken from the Inventory of Parent and Peer Attachment (IPPA) questionnaire [34]. It consists of 5 statements, which should be assessed in terms of truthfulness. The reliability of that scale in the examined sample was satisfactory (Cronbach’s alpha = 0.890). The scale of neighborhood social capital comprised of four statements on social relations in the neighborhood, i.e., social bonds and security; the responses were encoded to 0‒4 points. Cronbach’s alpha for this scale was 0.750. This scale, developed for the 2001/2002 HBSC study protocol by the Social Inequalities Focus Group, based on studies by Kawachi et al. [35], was tested in many countries, including Poland.

Among health-related behaviors, selected health-risk and pro-health behaviors were identified. Physical activity was examined using the MVPA (moderate-to-vigorous physical activity) indicator taken from Prochaska’s screening test [36]. It indicates the number of days in the preceding week when the respondent exercised with moderate intensity for a total of at least 60 minutes a week. The recommended level in the population of school youth was 7 days a week of moderate-to-vigorous physical activity. The MPVA scores were treated as a continuous scale.

Among eating habits, breakfast regularity and consumption of fruits and vegetables, as well as not recommended food items (sweets and sweetened beverages) were identified. Eating habits as variables were re-coded to 0–1 variable, where “1” indicated having breakfast every day, consuming fruits and vegetables every day, eating sweets and drinking sweetened beverages no more than once a week.

Assessment of health risk behaviors in the preceding 30 days included questions about tobacco smoking, alcohol consumption, getting drunk, and marijuana smoking. These questions were adapted for the HBSC protocol from the questionnaire of the European School Survey Project on Alcohol and Other Drugs [37]. The respondents indicated the frequency of identified behaviors on a seven-point scale (1–7), which was categorized in three groups of responses: never, 1‒2 times, and more often than 2 times in the last 30 days.

### 2.3. Statistical Analysis

The first step consisted of comparing crude mean PAS indexes in sub-groups of adolescents’ gender, age, and family affluence. The significance of differences between various sub-samples was examined with ANOVA. For the purposes of mean PAS index comparisons, continuous scales representing independent variables were categorized into three levels, and the average level was set based on about 50% of average cases. The calculation of correlations of variables among continuous scales was also part of the simple univariate analysis. The second step included an estimation of three linear regression models, in which the standardized PAS index was the dependent variable.

To explore the potential impact of behavioral factors adjusted for socio-demographic factors in subsequent models (steps), it was checked whether there is any more important predictor than behavioral and whether health behaviors remain significant. The factors were entered into the model in the following order: (1) the impact of health-related behaviors adjusted for demographic and socioeconomic variables, (2) school performance variables, (3) social relations variables. The general model was compared with models created for boys and girls. Standardized beta regression coefficients and the R^2^ determination coefficient were calculated to check how close the data were to the fitted regression equation. The statistical software IBM SPSS Statistics for Windows, v. 21.0. (IBM Corp., Armonk, NY, USA) was used for analysis.

## 3. Results

The raw PAS index had a range of 0–20 points. The mean score was 13.25 (SD = 3.74), which accounts for 66.3% of the maximum score. The mean score was 13.55 (SD = 3.62) for boys and 12.98 (SD = 3.81) for girls (*p* < 0.001). The mean PAS index was 13.36 (SD = 3.79) for 15-year-olds and 13.12 (SD = 3.67) for 17-year-olds (*p* = 0.073). Family affluence measured using the FAS correlated (although weakly) with the PAS; *r* = 0.077 (*p* < 0.001). The correlations between the variables are shown in Appendix A. Moreover, we found that gender-dependent differences were higher in older adolescents from cities and more affluent families.

Table 1 shows a comparison of the PAS indexes in groups differing in health-related behaviors and substance use.

Physically active adolescents who ate fruits and vegetables and breakfast every day demonstrated a more positive attitude than other peers. Consumption of sweets and soft drinks did not have a significant relationship with the PAS index level. Regarding all four variables relating to substance use, it was found that a clear link with the PAS exists. Adolescents who did not drink alcohol in the preceding 30 days scored higher. The weakest relationship was with cannabis use. The differences in the PAS scores regarding tobacco smoking and alcohol drinking were stronger in girls. In the case of incidents of getting drunk in the last 30 days, the difference was significant only in girls. In simple correlation analysis, the relation between PAS and substance use was statistically significant in all groups, and the results are shown in the Appendix A. Pearson’s coefficients were negative and relatively low: tobacco smoking *r* = −0.092, alcohol drinking *r* = −0.082, being drunk *r* = −0.070, cannabis smoking *r* = −0.050.

Table 2 shows a similar comparison of the PAS indexes in adolescents with regard to school performance and social factors.

The school performance factors included academic achievements and school stress. Social factors consisted of communication in the family and with peers and social capital in the neighborhood. Significant associations were noted in both genders (Appendix A). School stress negatively correlated with the PAS at a relatively high level (*r* = −0.249), and academic achievements were positively correlated (*r* = 0.278). Social relations variables were correlated with the PAS. Communication in the family at a high level was positively correlated (*r* = 0.366). For relation with peers, the correlation coefficient was *r* = 0.194, and the correlation coefficient for the social capital of the neighborhood was relatively high too (*r* = 0.232). The results of correlations also provided an opportunity to analyze the relationship between the positive attitude predictors. For example, attention was drawn to the relationship between school achievements and psychoactive substance use. Pearson’s correlation coefficients were significantly negative: tobacco smoking *r* = −0.250, alcohol consumption *r* = −0.205, being drunk *r* = −0.188, and cannabis use *r* = −0.135. Socio-demographic factors such as age, gender and FAS correlated with some other independent variable and were considered as confounders. Boys reported a significantly higher PAS score than girls, while the relationship with age and FAS was close to being significant (*p* = 0.07). It was concluded that the assumption for confounding was met.

Table 3 shows the results of the estimates of multiple linear regression models.

The first simple model with sociodemographic variables, health-related behaviors, and substance use explains only 5.7% of the PAS variability. It indicates a stronger influence of pro-healthy behaviors than harmful ones. Two variables referring to school performance significantly enhanced the quality of the adjustment of the second model (R^2^ = 0.128). After they were added, differences between boys and girls became apparent. The influence of physical activity and eating habits remained significant, though standardized regression coefficients were reduced. Given that the associations with substance abuse are not significant and that academic achievements and substance use were strongly correlated, it may be that the impact of risk behaviors on a positive attitude may be indirect. The relationship between alcohol drinking and the PAS level was close to the significance threshold (*p* = 0.053); after the additional explanatory variables had been added, it became insignificant.

The final model, encompassing communication in the family, relations with peers, and the neighborhood social capital, explains 25.1% of the PAS variability. Following the introduction of the new variables, the impact on family affluence loses its predictive power. However, it should be pointed out that the quality of communication in family correlated with family affluence (*r* = 0.172). Neighborhood social capital remained in the final model, despite the correlation with communication in the family (*r* = 0.252). Correlations are shown in Appendix A.

The findings indicate that substance use did not have a strong association with the PAS variability, but pro-health behaviors, such as physical activity and regular breakfasts, had a significant relationship. This is in contrast to the inverse correlation found between substance abuse, though this link appeared in simple comparisons of the mean indexes. Attention should be drawn to gender as a modifying factor for the examined associations. If models similar to those shown in Table 3 were estimated separately for girls and boys, then in girls, getting drunk remained the “positive attitude reducing” factor. Gender-specific models are shown in Table 4, where significant variables were included, which were listed in the order in which they were introduced to the model (stepwise method). Table 4 presents variables significant in at least one gender. In the final model for the total group, the relationship with eating breakfast was weak, and this predictor did not qualify, for either the boys nor the girls’ model. Gender turned out to be a factor triggering a relationship with the place of residence.

Changes in the R^2^ coefficient were identified following the introduction of each variable. In girls, the impact of the place of residence was visible—girls from rural areas scored higher in the PAS. The female-specific model explained PAS variability slightly better; the number of predictors was greater than for boys. Physical activity had less impact on the PAS variability in girls compared to boys. It was introduced to the model following the variable of being drunk. The influence of communication in the family was stronger in girls, but the impact of communication with peers and school stress was weaker in girls than in boys.

## 4. Discussion

The objective of this study was to identify social and behavioral determinants of adolescents’ positive attitudes towards themselves and their life. The short, validated Positive Attitude Scale (PAS), which is a 4-item scale, was applied. Similar attempts to develop positive thinking skills scales were undertaken by other researchers [38].

Our study showed that a moderate level of the possitive attitude was reported by Polish adolescents in the age group of 15–17 years. The mean score was 13.25 (SD = 3.74), which accounts for 66.3% of the maximum PAS score in a range of 0–20 points.

A positive attitude was associated with pro-health behaviors, mainly physical activity and eating breakfast while avoiding psychoactive substance use was not significant. Social factors such as communication in the family, relations with peers, and neighborhood social capital had an important role too. Moreover, good academic achievements and controlled levels of school stress were related to adolescents’ positive attitude.

With regard to social factors, communication in the family and relations with peers as well as the social capital of the neighborhood appeared to be strong predictors of a positive attitude. The extent of the contribution of these factors was different in boys and girls. The findings of the current study are in line with previous studies that demonstrated that positive self-concept, low depressive symptoms, and high perceived parental support and school connectedness were most strongly associated with adolescents’ optimism [39]. In addition, the study of Orejudo et al. [40] showed the important role of relations in the family and with peers in positive attitude development and differences regarding gender. For boys, having positive relations with peers was more strongly related to optimism than in girls, and for girls, family communication seemed to be a stronger predictor. The association of family factors with a positive attitude was examined in Polish adolescents, and communication in the family was proved to be the main predictor of adolescents’ optimism and subjective health [41]. 

Complex analysis of the influence of neighborhood environment on mental health with regard to optimism has been shown by Ruiz et al. [42]. They have proved that the association of neighborhood characteristics with optimism was significant, including observed social resources, as well as the perception of it. It is difficult to assume that individuals with a positive attitude towards themselves and the world better assess their neighborhood regardless of observed indicators. This implies that there is a relationship between observed neighborhood resources and residents’ perceptions. The observational studies suggest that neighborhood factors are associated with individuals’ overall self-rated subjective well-being [43]. In the study of Kleszczowska et al. [44], it was demonstrated that perception of the surrounding neighborhood proved to be a significant predictor of adolescents’ psychological well-being. Moreover, the protective impact of physical activity appeared to be stronger in less-supportive and low social capital neighborhoods.

School performance also had a strong influence on adolescents’ positive attitude. With regard to the four questions contained in the PAS, it may be noted that two of them referred to achievements and to coping skills of an individual (I am proud of my achievements; My belief in myself gets me through hard times in my life). The scale may be considered to measure satisfaction with performing developmental tasks, which are appropriate for the respondents’ age, such as school performance. The results showed a strong correlation between the self-assessment of school achievements and positive attitude. Adolescents who assessed their academic achievements higher had better results on the PAS, which leads one to believe that satisfaction with school performance is a starting point for positive self-perception of individual psychological resources. Such positive effects were shown in another study by Vanno et al. [45]. School-related stress had a negative impact on the positive attitude of both genders. A study by Huan et al. [46] also showed a negative relationship between adolescents’ optimism and perceived academic stress, but gender was not significant.

In the model describing adolescents’ positive attitude determinants, the influence of health risk behaviors was much weaker compared to health-enhancing behaviors. However, in a simple analysis of means of substance use, higher PAS indexes were obtained in those who practiced abstinence. In the multivariate analysis, alcohol abstinence remained in the group of independent predictors of positive attitude only in girls. However, attention should be drawn to the clear negative correlations between school achievements and psychoactive substance use, which is in line with the results of other studies [47]. The findings of this study showed a relatively weak but significant negative correlation between substance abuse and a positive attitude. Since school achievements came second in the final model it may be considered as an indirect effect of substance abuse on a positive attitude through school achievements.

Among pro-health behaviors, physical activity was one of the stronger factors enhancing adolescents’ positive attitude. In a monograph on the psychology of physical activity, Biddle and Mutrie [48] quoted the results of numerous studies confirming the relationship between physical activity and mental health. They proved that physical activity had a positive impact on mood; it affected psychological well-being and strengthened self-esteem, and it also had a positive effect on personality, self-regulation development, and general mental health [49]. The complex nature of the relationship between physical activity, self-esteem, and mental health was explained by the authors using sport enjoyment models [50] as well as Sonstroem’s model [51]. In the first case, sport enjoyment was analyzed in two intersecting continua: intrinsic vs. extrinsic group and achievements vs. non-achievements. The positive assessment of achievements confirm this study’s results, i.e., enhancement of positive attitude through physical activity. The intrinsic-achievement group showed a more positive internal assessment of their competences and successes owing to their physical activity. Conversely, a feeling of pride can be additionally intensified by social approval in the extrinsic-achievement group. Sontroem’s model comprises of two strands: the skills development hypothesis and the motivational approach. Optimism-related traits such as self-esteem, a feeling of success, competences to cope, and self-acceptance are assumed to be shaped and enhanced by physical activity. Conversely, positive self-esteem and optimism increase the motivation for further engagement in physical activity.

The recent HBSC survey conducted in 2017/18 in Polish adolescents show the complex relationships between behavioral and social factors and their impact on the variability of psychological well-being among adolescents [44]. Physical activity and perception of the surrounding social environment proved to be a significant predictor of youths’ mental health. The mental health-protective effect of physical activity appeared to be stronger and of more advantage in less supportive neighborhoods.

The limitations of this study come from the cross-sectional nature of this research, which does not allow causal factors to be identified. The same factors that have been identified as determinants of adolescents’ positive attitude could be stronger in optimistic people. Therefore, we should consider the reverse associations, which were proven in other studies [10,15,16,20,23,25]. However, the results of this study can be regarded as identification of factors that are associated with adolescents’ positive attitude, which in turn is related to optimism, an important indicator of mental health. Because of the time passed, the use of data from the HBSC survey conducted in 2010 could be considered as a limitation of this analysis. However, it seems that analyzed relationships and associations are universal. In the planning of large population surveys, it would be worthwhile to incorporate the PAS as a useful and short tool to assess positive attitude, which plays an important role in the development of adolescent optimism, by referring to individual factors considered in the Canadian model cited in the Introduction [7].

## 5. Conclusions

Adolescent optimism is associated with a wide variety of well-being and positive health outcomes, including mental and physical health, motivation to health-related lifestyle change, perception of social support, and school performance. Positive attitude towards life and self, in general, is an important cognitive component of optimism. Knowledge about the relationship between positive attitude and health behaviors, such as physical activity, social relations with family, peers, and neighbors, and school performance should be taken into consideration while designing health promotion programs for adolescents. Optimism is an important individual trait related to young people’s health and well-being, and a positive attitude, as its cognitive component, may be modified and developed in adolescence. Physical activity is an important factor influencing a positive attitude, along with positive social relations with family, peers, and neighbors, and school achievements. Based on the findings of this study, the increase in adolescents’ positive attitude could be expected after combining initiatives to promote physical activity and interpersonal communication competences development in pupils’ friendly school and supportive neighborhood environments.

## Figures and Tables

**Table 1 ijerph-16-04404-t001:** Mean crude Positive Attitude Scale (PAS) scores according to health-related behaviors by gender.

Health-Related Behaviors	*N* (%)	Boys			Girls		
		Mean PAS	SD	*p*	Mean PAS	SD	*p*
Physical activity (MVPA)							
0–4 days	1934 (64.4)	13.10	3.38		12.73	3.86	
5–6 days	676 (22.5)	13.68	3.51	<0.001	13.68	3.46	<0.001
7 days	394 (13.1)	14.54	4.13		13.93	3.77	
Breakfasts							
every day	1468 (49.9)	13.94	3.49	<0.001	13.43	3.57	<0.001
not every day	1471 (50.1)	13.01	3.73		12.62	3.94	
Fruits/vegetables							
every day	979 (33.4)	14.01	3.46	0.003	13.45	3.81	<0.001
not every day	1949 (66.6)	13.36	4.02		12.71	3.75	
Sweets/soft drinks							
no more than once/week	390 (13.2)	13.47	4.10	0.777	13.16	3.95	0.503
more often	2546 (86.8)	13.55	3.56		12.98	3.77	
Tobacco smoking in the last 30 days							
never	1934 (65.6)	13.72	3.58		13.32	3.65	
1–2 times	297 (10.1)	13.38	3.32	0.038	12.65	3.85	<0.001
more often	716 (24.3)	13.14	3.80		12.18	4.11	
Alcohol drinking in the last 30 days							
never	1084 (36.8)	13.78	3.59		13.63	3.76	
1–2 times	861 (29.2)	13.67	3.50	0.035	12.78	3.62	<0.001
more often	1002 (34.0)	13.21	3.71		12.35	3.96	
Been drunk in the last 30 days							
never	2191 (74.6)	13.63	3.57		13.25	3.72	
1–2 times	501 (17.1)	13.27	3.18	0.281	12.27	3.72	<0.001
more often	246 (8.4)	13.32	4.50		11.37	4.59	
Cannabis use in the last 30 days							
never	2633 (90.9)	13.65	3.60		13.02	3.77	
1–2 times	151 (5.2)	12.98	3.12	0.028	12.52	4.02	0.052
more often	111 (3.8)	12.70	4.15		11.57	4.74	

**Table 2 ijerph-16-04404-t002:** Mean crude Positive Attitude Scale (PAS) scores according to school performance and social relations by gender.

School Performance and Social Relations (Type/Range of Scale)	*N* (%)	Boys			Girls		
		Mean PAS	SD	*p*	Mean PAS	SD	*p*
School achievements (order)							
below average	240 (8.1)	11.22	4.49		10.27	4.54	
average	1415 (47.9)	13.22	3.48	<0.001	12.32	3.79	<0.001
good	919 (31.2)	14.13	3.41		13.63	3.41	
very good	377 (12.8)	15.46	3.62		14.77	3.27	
School stress (0–16)							
low	868 (29.4)	14.65	3.09		14.21	3.43	
average	1344 (45.6)	13.09	3.47	<0.001	12.95	3.64	<0.001
high	737 (25.0)	12.80	4.18		11.77	4.09	
Communication with family (0–55)							
low	723 (25.4)	12.25	4.20		10.97	4.09	
average	1427 (50.1)	13.40	3.15	<0.001	13.12	3.28	<0.001
high	699 (24.5)	15.41	3.02		14.77	3.39	
Communication with peers (0–20)							
low	729 (25.5)	12.58	3.75		11.68	4.23	
average	1371 (47.9)	13.73	3.20	<0.001	12.74	3.60	<0.001
high	763 (26.7)	15.25	3.49		13.90	3.64	
Neighborhood social capital (0–16)							
low	662 (23.7)	12.41	3.90		12.08	4.30	
average	1555 (55.7)	13.52	3.40	<0.001	12.91	3.54	<0.001
high	576 (20.6)	14.90	3.27		14.53	3.40	

**Table 3 ijerph-16-04404-t003:** Estimation of linear regression models—standardized beta values; dependent variable—standardized Positive Attitude Score (PAS).

Independent Variables (Type)	Model 1		Model 2		Model 3	
	Beta	*p*	Beta	*p*	Beta	*p*
Socio-demographic						
Gender (0–1)	0.035	0.085	**0.040**	**0.040**	**0.103**	**0.000**
Age (0–1)	0.020	0.304	0.038	0.052	0.036	0.051
Living in big city (0–1)	−0.039	0.091	−0.033	0.138	−0.013	0.547
Living in rural areas (0–1)	0.024	0.299	0.022	0.330	0.013	0.543
Family affluence (cont.)	**0.069**	**0.000**	**0.042**	**0.023**	−0.011	0.547
Health behaviors						
Physical activity (cont.)	**0.150**	**0.000**	**0.134**	**0.000**	**0.096**	**0.000**
Breakfast every day (0–1)	**0.097**	**0.000**	**0.074**	**0.000**	**0.040**	**0.024**
Often fruits and vegetables (0–1)	**0.059**	**0.002**	**0.047**	**0.012**	0.021	0.226
Rarely sweets and sweet drinks (0–1)	0.001	0.972	0.005	0.780	−0.009	0.603
Substances abuse in the past 30 days						
Tobacco smoking (cont.)	−0.011	0.629	0.033	0.137	0.030	0.159
Alcohol drinking (cont.)	−0.054	0.053	−0.040	0.135	−0.026	0.303
Being drunk (cont.)	−0.022	0.409	−0.011	0.654	−0.009	0.699
Cannabis use (cont.)	−0.020	0.324	−0.018	0.376	−0.015	0.434
School performance						
Academic achievements (cont.)	-	-	**0.184**	**0.000**	**0.170**	**0.000**
School-related stress (cont.)	-	-	**−0.159**	**0.000**	**−0.110**	**0.000**
Social relations						
Communication in the family (cont.)	-	-	-	-	**0.253**	**0.000**
Communication with peers (cont.)	-	-	-	-	**0.164**	**0.000**
Neighborhood social capital (cont.)	-	-	-	-	**0.100**	**0.000**
Adjusted R^2^	0.057		0.128		0.251	

Significant results are bolded

**Table 4 ijerph-16-04404-t004:** Estimation gender-specific regression models with stepwise selection; dependent variable—standardized Positive Attitude Score (PAS).

Independent Variables (Type) in the Order of Entry	Boys			Girls		
	Beta	*p*	Change R^2^	Beta	*p*	Change R^2^
Communication in the family (cont.)	0.221	0.000	0.121	0.285	0.000	0.158
Academic achievements (cont.)	0.168	0.000	0.046	0.169	0.000	0.044
Communication with peers (cont.)	0.175	0.000	0.037	0.144	0.000	0.023
Neighborhood social capital (cont.)	0.110	0.000	0.013	0.094	0.000	0.012
School-related stress (cont.)	−0.127	0.000	0.012	–0.095	0.000	0.009
Physical activity (cont.)	0.110	0.000	0.012	0.074	0.002	0.007
Being drunk (cont.) *	-	-	-	–0.102	0.000	0.005
Living in rural areas (0–1)	-	-	-	0.059	0.014	0.003
Adjusted R^2^		0.241			0.261	

* In girls, “Being drunk” entered the final model before “Physical activity”.

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
