# Peer review of "Social and Behavioral Predictors of Adolescents’ Positive Attitude towards Life and Self"

_ijerph, 2019, doi:10.3390/ijerph16224404_

Round 1
Reviewer 1 Report
This manuscript describes the main predictors of the positive attitude towards life and self in adolescence. The role of optimism in adolescents’ health, which is related to positive attitude, has been widely studied. However, there is limited knowledge about the social and behavioral factors contributing to the positive attitude development. The authors performed set of statistical analyses using data collected from 15- to 17-year-old adolescents with the dependent variable being the Positive Attitude Scale score. Univariate ANOVA revealed significant differences in crude mean PAS score between sub-groups of several health-related behaviors, school performance and social relations in girls and boys separately. Then hierarchical linear regression models were applied using the standardized PAS score as the dependent variable. Eight out of 18 variables were significant in the final model, which explained 25.1% of PAS variability. Among these 8 variables, communication in the family and with peers as well as neighborhood social capital showed the strongest impact on positive attitude in adolescents. Physical activity, eating breakfast and school performance were also found to be important predictors but to a lesser degree. Finally, the contribution of most of these 8 factors was tested using stepwise regression models in boys and girls separately. The female-specific model explained PAS variability slightly better as the number of predictors was greater than for boys. Therefore, the results of the study highlight the need to include social relations and health-related behaviors in mental health promotion programs, particularly in girls. The analyses performed in this study are suitable to address the question which factors contribute the most to the positive attitude of adolescents in a gender-specific manner. The topic of the study is of interest, however, as optimism and positive attitude are related, the contribution of the findings of the current study to the existing literature regarding the relationship between optimism and mental health and factors predicting optimism needs to be clarified.
Abstract
Lines 16-17: The link between the two sentences is missing. It should be clarified why it is important to study positive attitude if optimism, which is related to positive attitude, has been widely explored. Line 22: Did you mean the ANOVA was adjusted? In the regression models the contribution of gender, age and family affluence was tested. Line 23: Could it be that there are 18 variables in the final model and not 17 as written?
Introduction
Lines or 53-55: “This study investigates associations and influence of social relations and health behaviors on adolescents’ positive attitude …” This should be rephrased to state that previous studies focused mainly on optimism, which is related to positive attitude, and then outline the findings of these studies. Line 64-65: “… positive attributes and optimism are inversely associated with…” “inversely” does not suit here as there probably is a positive relationship between positive attributes and optimism and physical health. The same for “risk” which probably cannot be determined by positive attributes and optimism. Line 66: “Optimism partially mediates…” it should be clarified what relationship (i.e. association) is mediated: relationships with parents, peers and teachers with what? Paragraphs 3, 4 and 5 are dedicated mainly to previous studies exploring optimism. It should be explained why it is important to study positive attitude in terms of novelty, how it is different, what it will add to the literature etc. In paragraphs 3, 4 and 5 it will be easier to understand which factor each paragraph refers to by starting with a sentence such as “ In regard to health behaviors… ”. Lines 77-78: Please explain why is it important to incorporate all these factors together in the analysis in comparison to previous studies, in other wards what is the novelty of this study?
Methods
Line 168: “… adolescents’ sub-groups” it should be added sub-groups of what. Lines 174-175: Please explain why the socio-demographic variables are entered into the models as shown in Table 3 if they are not tested as potential predictors for the positive attitude. In addition, if these factors were meant to be in the models why the health behaviors were adjusted for these variables? Lines 176-177: It needs to be added that these are stepwise models and justify why they are used, i.e. what is the rational behind using the stepwise model. The type of the stepwise model should be specified as well.
Results
Lines 184-185: Where these findings are shown? Table 1: Physical activity: percentages exceed 100. Line 216: These are multiple linear regression models and not multivariate models as there is only one outcome and several predictors. This should be changed throughout the manuscript. Lines 226-228: The sentence should be changed to include the result of the linear regression model “Given that the associations with substance abuse are not significant and that academic achievements…. substance use, it may be that the impact of risk behaviours…” Lines 235-236: This is unclear as neighborhood social capital was added last to the model. Lines 243-245: Please explain why eating breakfast was not introduced into the girls’ model if found to be significant in the final general model. Also, why living in rural areas was introduced to both models if came out non-significant in the final general model?
Discussion
Line 305-307: To be more accurate here this should be considered as an indirect effect of substance abuse on positive attitude through school achievements and not a mediation. In order to mediation to occur there should have been a correlation between substance abuse and positive attitude. Lines 337-338: Please explain these sentences. Did you mean because of the time passed since 2010? “However, it seems that …. are universal.” Did you mean this because other studies show similar results? Lines 341-342: It should be stated here what is the value in adding this scale, how is this different from using optimism scale?
Conclusion
Line 343: “Positive attitude towards … and well-being.” This was not explored in the present study and there were no citations of articles pointing to this relationship. It could be implied from other traits such as optimism but then this should be re-phrased to be stated more accurately. Line 344: “… can be considered as individual factor…” the word factor should be replaced with another word such as trait so not to confuse with the factors found to have a contribution to positive attitude (i.e. the determinants in this sentence). Lines 349-350: “…combining initiatives to promote physical activity…“ this sentence should come before the sentence starting in line 345 and the focus of the sentence in lines-345-348 should then be on the positive attitude instead of optimism. Lines 350-353: The focus of this sentence should be that it is essential to develop initiatives that will promote physical activity and communication.
Supplementary data
Reference to these data should be made throughout the manuscript where simple correlations are mentioned.
Language
Proofreading should be done to correct minor errors in grammar and typos including in the tables titles. Some of these are listed here:
Line 18: Replace purpose with aim. Line 19: Replace toward with towards throughout the manuscript. Line 21: Replace 4-items with 4-item. Line 25: Add “as well as” before neighbourhood and change “impact to” to “impact on”. Line 27: The sentence should begin with “the results of the current study highlight”. Line 40: “… positive development including” did you mean “and includes”? Line 53: “…. health behaviors on adolescents’ positive attitude” “on” should be replaced with “with”. Line 63: … by comparison to health behaviors” did you mean “in comparison with”? Line 169: “…significance of differences between various factors” the differences are between sub-samples not factors. Line 177: “…respectively.” is not needed. Line 237: Replace “conducted analyses” with “findings”. Lines 239-240: This sentence should be “This is in contrast to the inverse correlation found between substance abuse…” Line 259: “… Positive Attitude Scale (PAS) - was applied, which was…” this sentence should be re-written as “… Positive Attitude Scale (PAS), which was…scales, was applied.” Line 280: The word “persons” should be replaced with “individuals” and “oneself” with “themselves”. Lines 268-269: “… strong predictors, and their role differed according to gender.” This should be re-phrased so it will be clear that the extent of the contribution of these factors was different in boys and girls. Lines 269: A sentence should be added to state “the findings of the current study are in line with previous studies which demonstrated….” Lines 281-282: “There is the…” if this sentence is linked to the previous one it should say something such as “this implies that…” Line 301: Please add much weaker compared to what. Line 350: “Reduction of negative and…” add emotions after negative. Also please change “of” to “in”.
Author Response
Dear Reviewer 1
We appreciate very much your valuable and detailed comments and suggestions to improve the quality of our manuscript. Thank you very much for your attention and effort in writing the review.
Below please find our responses to Your comments with our explanations point-by-point with the details of the revisions in the manuscript
Response to Reviewer 1 Comments
Point 1 (Abstract): Lines 16-17: The link between the two sentences is missing. It should be clarified why it is important to study positive attitude if optimism, which is related to positive attitude, has been widely explored.
Response 1: We changed the first sentences to clarify why it is important to study predictors of positive attitude. According to the dispositional optimism definition positive attitude is important cognitive component of the optimism.
Point 2 (Abstract): Line 22: Did you mean the ANOVA was adjusted? In the regression models the contribution of gender, age and family affluence was tested.
Response 2: We corrected the sentence about conducted analysis of univariate ANOVA and hierarchical linear regression models.
Point 3 (Abstract): Line 23: Could it be that there are 18 variables in the final model and not 17 as written?
Response 3: Eight out of 18 variables were included in the final model, so we corrected it. In Abstract regarding the comments we also rephrased the last sentence.
Point 4 (Introduction): Lines or 53-55: “This study investigates associations and influence of social relations and health behaviors on adolescents’ positive attitude …” This should be rephrased to state that previous studies focused mainly on optimism, which is related to positive attitude, and then outline the findings of these studies.
Response 4: We rephrased the sentence to state that previous studies focused mainly on optimism, which is related to positive attitude, and then we outlined the findings of these studies.
Point 5 (Introduction): Line 64-65: “… positive attributes and optimism are inversely associated with…” “inversely” does not suit here as there probably is a positive relationship between positive attributes and optimism and physical health. The same for “risk” which probably cannot be determined by positive attributes and optimism.
Response 5: We changed the sentence and we wrote about positive correlations between positive attributes and optimism with physical health, and risk reduction of metabolic and cardiometabolic diseases in youth.
Point 6 (Introduction): Line 66: “Optimism partially mediates…” it should be clarified what relationship (i.e. association) is mediated: relationships with parents, peers and teachers with what? Paragraphs 3, 4 and 5 are dedicated mainly to previous studies exploring optimism. It should be explained why it is important to study positive attitude in terms of novelty, how it is different, what it will add to the literature etc. In paragraphs 3, 4 and 5 it will be easier to understand which factor each paragraph refers to by starting with a sentence such as “ In regard to health behaviors… ”.
Response 6: We clarified that dispositional optimism mediates adolescents’ perceived support from parents, peers, and teachers to psychological well-being. We changed paragraphs accordingly to the suggestions to make easier to understand which factor each paragraph refers to, by starting with a sentence such as “ In regard to … ”.
Point 7 (Introduction): Lines 77-78: Please explain why is it important to incorporate all these factors together in the analysis in comparison to previous studies, in other wards what is the novelty of this study?
Response 7: We explained that, in comparison to previous studies about youth’s optimism, it seems to be important and interesting to incorporate in the analysis social relations, health behaviors and school performance to exam the association to positive attitude, since the positive attitude significantly contributes to optimism development in adolescence.
Point 8 (Methods): Line 168: “… adolescents’ sub-groups” it should be added sub-groups of what.
Response 8: We added that the first step consisted of comparing crude mean PAS indexes in adolescents’ sub-groups of gender, age and family affluence.
Point 9 (Methods): Lines 174-175: Please explain why the socio-demographic variables are entered into the models as shown in Table 3 if they are not tested as potential predictors for the positive attitude. In addition, if these factors were meant to be in the models why the health behaviors were adjusted for these variables?
Response 9: We considered socio-demographic factors as confounders. Age, gender and FAS strongly correlated with some other independent variables. Boys reported significantly higher PAS score than girls, while the relationship with age and FAS was close to be significant (p=0.07). It could therefore be concluded that the assumption for confounding was met.
Point 10 (Methods): Lines 176-177: It needs to be added that these are stepwise models and justify why they are used, i.e. what is the rationale behind using the stepwise model. The type of the stepwise model should be specified as well.
Response 10: First of all we wanted to explore the potential impact of behavioral factors adjusted for socio-demographic factors. In subsequent models (steps) we checked whether there is any more important predictor than behavioral and whether health behaviors remain significant.
Point 11 (Results): Lines 184-185: Where these findings are shown?
Response 11: We added that the findings of correlations are shown in Table A1 in Supplementary data.
Point 12 (Results): Table 1: Physical activity: percentages exceed 100.
Response 12: We corrected in Table 1 physical activity percentages.
Point 13 (Results): Line 216: These are multiple linear regression models and not multivariate models as there is only one outcome and several predictors. This should be changed throughout the manuscript.
Response 13: It was one dependent variable. It has been changed and corrected in the manuscript for “multiple linear regression models”.
Point 14 (Results): Lines 226-228: The sentence should be changed to include the result of the linear regression model “Given that the associations with substance abuse are not significant and that academic achievements…. substance use, it may be that the impact of risk behaviours…”
Response 14: We changed the sentence.
Point 15 (Results): Lines 235-236: This is unclear as neighborhood social capital was added last to the model.
Response 15: The last block of independent variables concerns social relations, including family, peers and neighbors. The scale of social capital was described in lines 145-149. To clarify we changed the line 176 and used the same term as in table 3 (variables related to social relations instead of social factors variables).
Point 16 (Results): Lines 243-245: Please explain why eating breakfast was not introduced into the girls’ model if found to be significant in the final general model. Also, why living in rural areas was introduced to both models if came out non-significant in the final general model?
Response 16: The total model (Table 3) included all variables under study, regardless the results turned out to be significant. We added that in the Table 4 we presented only variables significant in at least one gender. In the final model for the total group the relationship with eating breakfast was weak (p=0.024) and this predictor did not qualify neither for boys nor girls model. In turn, gender turned out to be a factor triggering a relationship with the place of residence.
Point 17 (Discussion): Line 305-307: To be more accurate here this should be considered as an indirect effect of substance abuse on positive attitude through school achievements and not a mediation. In order to mediation to occur there should have been a correlation between substance abuse and positive attitude.
Response 17: We corrected this sentence, and we wrote that the findings of this study showed relatively weak but significant negative correlations between substance abuse and positive attitude, so it may be also considered as an indirect effect of substance abuse on positive attitude through school achievements.
Point 18 (Discussion): Lines 337-338: Please explain these sentences. Did you mean because of the time passed since 2010? “However, it seems that …. are universal.” Did you mean this because other studies show similar results?
Response 18: We meant this because of time passed since 2010.
Point 19 (Discussion): Lines 341-342: It should be stated here what is the value in adding this scale, how is this different from using optimism scale?
Response 19: We explained what is the value in using PAS to measure predictors of optimism since this short item scale assess the positive attitude which plays important role in development of optimism in adolescence.
Point 20 (Conclusions): Line 343: “Positive attitude towards … and well-being.” This was not explored in the present study and there were no citations of articles pointing to this relationship. It could be implied from other traits such as optimism but then this should be re-phrased to be stated more accurately. Line 344: “… can be considered as individual factor…” the word factor should be replaced with another word such as trait so not to confuse with the factors found to have a contribution to positive attitude (i.e. the determinants in this sentence). Lines 349-350: “…combining initiatives to promote physical activity…“ this sentence should come before the sentence starting in line 345 and the focus of the sentence in lines-345-348 should then be on the positive attitude instead of optimism. Lines 350-353: The focus of this sentence should be that it is essential to develop initiatives that will promote physical activity and communication.
Response 20: We rephrased the whole part of conclusions according to the comments and suggestions.
Point 21 (Supplementary data): Reference to these data should be made throughout the manuscript where simple correlations are mentioned.
Response 21: We included throughout the manuscript the references to Supplementary data, Table S1 with correlations results.
Point 22 (Language): Proofreading should be done to correct minor errors in grammar and typos including in the tables titles.
Response 22: We did proofreading of the text and we corrected errors in grammar and typos including tables.

Reviewer 2 Report
The work entitled “Social and behavioral predictors of adolescents’ 2 positive attitude towards life and self.” contains new scientific knowledge and covers a relevant topic. However, I have some comments to make that should be addressed before the manuscript could be considered for publication.
- I have some concerns about the introduction and the background information. It seems that the study is about adolescents’ mental health but no information can be found about previous studies analyzing this aspect. More studies about adolescent wellbeing and mental health should be included in order to add information to further comprehend this aspect and to enhance the comparison of the results found in the present study (Chen and Harris, 2019; Fonseca-Pedrero, Inchausti et al., 2018; Ortuño-Sierra, Aritio-Solana, Fonseca-Pedrero, 2018; Steinvoord and Junge, 2019)
-Authors mentioned that the sample is representative, what do not explain how. Was the sample randomly selected? If so they should explain how, if it was incidentally selected, this should also be explained. Also, what was the gender distribution of each age subsample and the then for the final sample?
- In the instruments subsection, authors should include citations about all the instruments used in the study (e.g. the FAS).
- What statistical program did the authors use for the study?
- Also, with regards to the statistical analysis, I think that a MANOVA should be more appropriate. Also, why authors did not include age as a factor in addition of gender?
- In the discussion section, authors sustain that “Our study showed that a moderate positive attitude was prevalent among Polish adolescents in the age group of 15-17 years”. What is the meaning of that affirmation? How was the attitude positive? Compare to what group?
- Finally, authors should consider revising the writing when talking about the study of internal consistency or convergent validity in some parts of the paper. First of all, validity is not a property of the test but inferences of the scores, and also, there are sources or validity evidences, as it is reflected in the APA standards. Attending to this approach it would be more appropriate to talk about evidences of internal structure or evidences of relation with other variables or external variables. In addition, the reliability is not a characteristic of the test. It is more correct to talk about reliability of the scores or estimation of the reliability of the scores (Prieto & Delgado, 2010).
Author Response
Dear Reviewer 2
We appreciate very much your valuable and detailed comments and suggestions to improve the quality of our manuscript. Thank you very much for your attention and effort in writing the review.
Below please find our responses to Your comments with our explanations point-by-point with the details of the revisions in the manuscript.
Response to Reviewer 2 Comments
Point 1: I have some concerns about the introduction and the background information. It seems that the study is about adolescents’ mental health but no information can be found about previous studies analyzing this aspect. More studies about adolescent wellbeing and mental health should be included in order to add information to further comprehend this aspect and to enhance the comparison of the results found in the present study (Chen and Harris, 2019; Fonseca-Pedrero, Inchausti et al., 2018; Ortuño-Sierra, Aritio-Solana, Fonseca-Pedrero, 2018; Steinvoord and Junge, 2019)
Response 1: In the Introduction we added more previous studies about adolescents’ mental health and well-being. We reviewed references suggested by reviewers and we selected two, which are valuable and are added to References:
[4] Chen P, Harris KM. Association of Positive Family Relationships With Mental Health Trajectories From Adolescence To Midlife. JAMA Pediatr. Published online October 07, 2019. doi:10.1001/jamapediatrics.2019.3336
[5] Ortuño-Sierra, J.; Aritio-Solana,R.; Fonseca-Pedrero,E.Mental health difficulties in children and adolescents: The study of the SDQ in the Spanish National Health Survey 2011-2012.Psychiatry Res. 2017 Oct 18; 259: 236–242.
Published online 2017 Oct 18. doi: 10.1016/j.psychres.2017.10.025
The longitudinal study of Chen and Harris [2019] showed that positive family relationships are associated with better mental health and reduced depressive symptoms among females and males from early adolescence to midlife
The findings of cross-sectional study of Steinvoord and Junge [2019] showed relationship between family socio-economic status and physical as well as social well-being German young adolescents. However, since these studies do not concern adolescents’ optimism or positive attitude directly, it was difficult for us to compere the results of our study to the findings of above mentioned previous studies in Discussion part.
Point 2: Authors mentioned that the sample is representative, what do not explain how. Was the sample randomly selected? If so they should explain how, if it was incidentally selected, this should also be explained. Also, what was the gender distribution of each age subsample and the then for the final sample?
Response 2: In Methods we added and explain the sampling procedure.
Sample selection and organization of the school-based survey was done according to the international research protocol, which is explained in reference [25]: Currie, C.; Griebler, R.; Inchley, J.; Theunissen, A.; Molcho, M.; Samdal, O.; Dür, W. (eds.). Health Behaviour in School-aged Children (HBSC) study protocol: background, methodology and mandatory items for the 2009/10 survey. Edinburgh: CAHRU & Vienna: LBIHPR. 2010, Available online: http://www.hbsc.org (accessed on 29 August 2019).
In Poland the HBSC national survey cluster sampling of school classes was carried out. The primary sampling unit were schools, a single class was chosen to be included in the sample. All pupils within selected classes were included in the sample. The procedure is explained in reference [26]: Mazur, J.; Małkowska-Szkutnik, A. [Results of Polish HBSC survey 2010. Technical report.]. Health Institute of Mother and Child, Warsaw, Poland 2011; pp. 5-18, ISBN 978-83-88767-58-6 (in Polish).
Point 3: In the instruments subsection, authors should include citations about all the instruments used in the study (e.g. the FAS).
Response 3: We included citation regarding instrument to measure family affluence , i.e. Family Affluence Scale (FAS) in reference [30] Currie, C.; Molcho, M.; Boyce, W.; Holstein, B.; Torsheim, T.; Richter, M. Researching health inequalities in adolescents: the development of the Health Behaviour in School-Aged Children (HBSC) family affluence scale. Soc Sci Med. 2008, 66(6), 1429–1436. doi:10.1016/j.socscimed.2007.11.024
Point 4:What statistical program did the authors use for the study?
Response 4: We added in the end of the part 2.3. Statistical Analysis that the statistical software IBM SPSS Statistics for Windows, v. 21.0. (IBM Corp., Armonk, NY, USA) was used for analysis.
Point 5: Also, with regards to the statistical analysis, I think that a MANOVA should be more appropriate. Also, why authors did not include age as a factor in addition of gender?
Response 5: To compare crude PAS scores univariate ANOVA analysis was conducted. Boys reported significantly higher score in Positive Attitude Scale than girls. The difference in PAS according to age was not significant (p=0.07). Therefore we decided not to include age in comparison of PAS indexes. However it was included in the final model.
Point 6: In the discussion section, authors sustain that “Our study showed that a moderate positive attitude was prevalent among Polish adolescents in the age group of 15-17 years”. What is the meaning of that affirmation? How was the attitude positive? Compare to what group?
Response 6: In Discussion we corrected the sentence and we explained that our study showed that a moderate level of the positive attitude scale was reported by Polish adolescents in the age group of 15-17 years. The mean score was 13.25 (SD=3.74), which accounts for 66.3% of the maximum PAS score in a range of 0-20 points.
Point 7: Finally, authors should consider revising the writing when talking about the study of internal consistency or convergent validity in some parts of the paper. First of all, validity is not a property of the test but inferences of the scores, and also, there are sources or validity evidences, as it is reflected in the APA standards. Attending to this approach it would be more appropriate to talk about evidences of internal structure or evidences of relation with other variables or external variables. In addition, the reliability is not a characteristic of the test. It is more correct to talk about reliability of the scores or estimation of the reliability of the scores (Prieto & Delgado, 2010).
Response 7: In Instruments we revised the sentence about internal consistency. Presenting PAS scales and other instruments we wrote about the reliability confirmed and evaluated using Cronbach alpha. We corrected one sentence with term “internal consistency”, and we changed it for “reliability” to use the same terminology. We did not use the term “convergent validity” anywhere in the text, because the analysis of validity of PAS was not included in the purpose of our study.

Round 2
Reviewer 2 Report
Authors have addressed all the points raised.
Nonetheless, there is a mistake with one citation.
Studies among German young adolescents found relationship between family socio-economic status and physical and social well-being (5). Please modify the sentence as this citation is related to spanish adolescents.
Author Response
Dear Reviewer 2
We appreciate very much your quick response and remark concerning the reference [5].
Response to Reviewer 2 Comments
Point 1: Studies among German young adolescents found relationship between family socio-economic status and physical and social well-being (5). Please modify the sentence as this citation is related to spanish adolescents.
Response 1: Thank you very much for your comment. The study of Ortuño-Sierra et al. [5] regarding assessment of mental health in children and adolescents were conducted among Spanish adolescents. We modified and corrected the sentence in Line 45.